# TETA: TEMPORAL-ENHANCED TEXT-TO-AUDIO GENERATION

## ABSTRACT

Large diffusion models have been successful in text-to-audio (T2A) synthesis tasks, but they often suffer from common issues such as semantic misalignment and poor temporal consistency due to limited natural language understanding and data scarcity. Additionally, 2D spatial structures widely used in T2A works lead to unsatisfactory audio quality when generating variable-length audio samples since they do not adequately prioritize temporal information. To address these challenges, we propose TETA, a latent diffusion-based T2A method. Our approach includes several techniques to improve semantic alignment and temporal consistency: Firstly, we use pre-trained large language models (LLMs) to parse the text into structured <event & order> pairs for better temporal information capture. We also introduce another structured-text encoder to aid in learning semantic alignment during the diffusion denoising process. To improve the performance of variable length generation and enhance the temporal information extraction, we design a feed-forward Transformer-based diffusion denoiser. Finally, we use LLMs to augment and transform a large amount of audio-label data into audio-text datasets to alleviate the problem of scarcity of temporal data. Extensive experiments show that our method outperforms baseline models in both objective and subjective metrics, and achieves significant gains in temporal information understanding, semantic consistency, and sound quality. Our demos are available at `https://teta2023.github.io/`.

## 1 INTRODUCTION

Deep generative learning models Goodfellow et al. (2020); Kingma & Dhariwal (2018); Ho et al. (2020) have revolutionized the creation of digital content, enabling creators with no professional training to produce high-quality images Rombach et al. (2022); Saharia et al. (2022); Nichol et al. (2021), vivid videos Hong et al. (2022); Singer et al. (2022), diverse styles of voice Huang et al. (2022), and meaningful long textual spans Zhang et al. (2022); OpenAI (2023). Professional practitioners can modify the generated content to accelerate their production workflows. Text-to-audio synthesis (T2A) is a subcategory of generative tasks that aims to generate natural and accurate audio by taking text prompts as input. T2A can be useful in generating desired sound effects, music or speech, and can be applied to various applications like movie sound effects making, virtual reality, game development, and audio editing.

Thanks to the development of text-to-image synthesis (T2I) methods, researchers have successfully extended similar approaches to the text-to-audio synthesis domain Huang et al. (2023a); Liu et al. (2023); Yang et al. (2023); Kreuk et al. (2023). The success of these methods has opened up numerous opportunities for generating high-quality audio content from text. T2A systems typically use a text encoder to encode the audio's text input as condition embedding, then employ diffusion models Huang et al. (2023a); Liu et al. (2023); Yang et al. (2023) to synthesis mel-spectrograms, or utilize autoregressive models Kreuk et al. (2023) to synthesis raw waveform data based on the condition embedding. However, previous T2A methods have some common issues: 1) **Temporal disorder**: when the text input is complex, with multiple objects and temporal relationships between them, the generated audios often suffer from semantic misalignment and temporal disorder. For instance, audio captions such as "The sound of A, followed by the sound of B" may result in audios where A and B overlapping throughout, or B comes before A, or even only one sound is synthesized. 2) **Poor variable-length results**: previous works Huang et al. (2023a) adopt the conventional U-Net

structure of 2D convolution and spatial transformer stacking as the backbone of diffusion denoiser, which is typically trained with fixed-length audios. Consequently, they generate suboptimal results when synthesizing audio sequences of varying lengths compared to those of the training data. In the meanwhile, 2D spatial structures are not good at extracting temporal information since they treat the time axis and frequency axis equally in the spectrogram generation process. 3) **Insufficient temporal paired data**: previous works use simple rule-based augmentation methods Elizalde et al. (2022); Kreuk et al. (2023) to create temporally aligned text-audio paired data from audio-label datasets. However, these patterns are overly simplistic and can hinder the model's ability to generalize to real-world sentences.

In this paper, we propose a novel temporal-enhanced text-to-audio generation framework. The temporal information can be better handled by our method in the following ways: 1) To address the semantic misalignment and temporal disorder, we use a pre-trained LLM to extract the audio caption's temporal information and parse the origin caption into structured <event & order> pairs with proper prompts. To encode the structured pairs better, we introduce another structured-text encoder that takes the structured pairs as its input to aid in learning semantic alignment during the diffusion denoising process. In this way, we relieve the text encoder's burden of recognizing events with the corresponding temporal information and enable the T2A system to model the timing information of the events more effectively. 2) To improve the generation quality of variable-length audio and enhance the temporal information understanding, we replace the 2D spatial structures with temporal feed-forward Transformer Ren et al. (2019) and 1D-convolution stacks for the diffusion denoiser and support variable-length audio input in training. 3) To address the issue of insufficient temporally aligned audio-text paired dataset, we use single-labeled audio samples and their labels to compose complex audio and structured captions. We then use LLM to augment the structured caption into natural language captions.

We conduct extensive experiments on AudioCaps and Clotho datasets, which reveals that our method surpasses baseline models in both objective and subjective metrics, and achieves significant gains in understanding temporal information, maintaining semantic consistency, and enhancing sound quality. Our ablation studies further demonstrate the effectiveness of each of our techniques.

## 2 RELATED WORKS

### 2.1 TEXT-TO-IMAGE GENERATIVE MODELS

Text-to-Image Synthesis (T2I) has garnered significant attention in recent years and has even been commercialized. One pioneering work in this realm is DALL-E Ramesh et al. (2021), which treats T2I generation as a sequence-to-sequence translation task. DALL-E employs a pre-trained VQ-VAE Van Den Oord et al. (2017) to encode image patches to discrete codes, which are then combined with the text codes. During inference, the model generates image codes autoregressively based on the text codes. As diffusion models exhibit greater potential with regard to both diversity and quality in image generation, they have become mainstream in T2I Synthesis. DALLE-2 Ramesh et al. (2022) uses the CLIP Radford et al. (2021) text encoder and two diffusion models. The first diffusion model predicts CLIP visual features based on the CLIP text feature, while the second synthesizes the image from the predicted CLIP visual features. A cascade of diffusion super-resolution models is then employed to increase the resolution of the generated image. Another famous T2I work is Imagen Saharia et al. (2022), which utilizes the T5 encoder Raffel et al. (2020) to extract text features, It employs a diffusion model to synthesize a low-resolution image and then applies a cascade of diffusion models for super-resolution. Latent Diffusion Rombach et al. (2022) enhances computational efficiency by using a continuous VAE that is trained with a discriminator to map images from pixel space to compressed latent space. This is followed by diffusion on the latent space, which synthesizes images' latent.

### 2.2 TEXT-TO-AUDIO SYNTHESIS

Text-to-Audio Synthesis is a rising task that has seen great advances recently. Diffsound Yang et al. (2023) uses a pre-trained VQ-VAE Van Den Oord et al. (2017) trained on mel-spectrograms to convert audio into discrete codes, which are then used by a diffusion model to generate the audio codes. To improve its generalization ability, the authors pre-trained Diffsound on the AudioSet

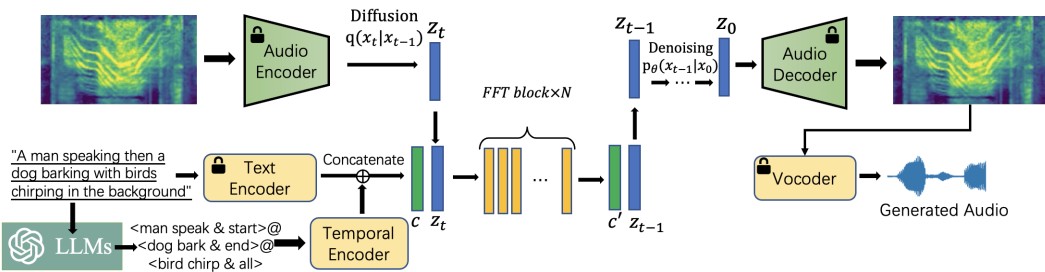

Figure 1: A high-level overview of our method. Note that modules printed with a *lock* are frozen when training the T2A model.

dataset, which contains audio files labeled with tags. Additionally, they introduce a random input masking technique to make use of these tags. AudioGen Kreuk et al. (2023) is another system in this field that uses a similar VQ-VAE-based approach. However, it encodes raw waveform data into discrete codes and employs an autoregressive model to predict audio tokens based on text features. For data augmentation, AudioGen mixes audio files and concatenates their text captions. Make-An-Audio Huang et al. (2023a), AudioLDM Liu et al. (2023), and TANGO Ghosal et al. (2023) are all based on the Latent Diffusion Model (LDM). With the assumption that CLAP can map the audio and its caption to the same latent space and approximate the text features based on the audio feature, AudioLDM uses audio features extracted by the CLAP model as the condition during training and utilizes text features during inference. In contrast, Make-An-Audio and TANGO employ text features both in the training and inference stages. To overcome data scarcity, Make-An-Audio proposes a pseudo prompt enhancement method, while TANGO introduces an audio mixture method based on human auditory perception.

### 2.3 LLM-BASED DATA AUGMENTATION

Recent advancements in prompt learning Liu et al. (2021) have greatly enhanced the capabilities of language models and given birth to very large language models with billions of parameters, enabling them to achieve natural language comprehension levels that are comparable to those of humans. OPT Zhang et al. (2022), ChatGPT[1] and GPT4 OpenAI (2023), are typical cases among them. This has led researchers to explore whether these models can be used to annotate data instead of humans. In previous work, masked language models (MLM) like BERT Devlin et al. (2018) and Roberta Liu et al. (2019) have been used for contextual augmentation Kumar et al. (2021) at the word level. For example, researchers insert <mask> tokens into the text or replace some words with <mask> tokens, and then use the MLM to predict the appropriate words. At the sentence level, back translation Sennrich et al. (2015) and paraphrasing Kumar et al. (2019) methods have been used to increase the diversity of data. However, the limited capabilities of these models result in insufficient quality and diversity of the generated data. To address these limitations, recent research has explored the use of very large language models for data augmentation. AugGPT Dai et al. (2023), for instance, leverages ChatGPT to generate auxiliary samples for few-shot text classification. The quality of the generated data is much higher, resulting in double-digit improvements in sentence classification accuracy compared to previous data augmentation methods. WavCaps Mei et al. (2023) crawls audio data with raw descriptions from multiple web sources. However, the raw descriptions contain a high degree of noise, so ChatGPT is used to filter out extraneous information unrelated to the audio and generate high-quality captions based on labels. This approach results in a large-scale, weakly-labeled audio captioning dataset.

## 3 METHOD

In this section, we begin by providing an overview of the framework of our method. We then introduce our temporal enhancement method and dual text encoder structure, which aims to capture

---

[1]https://openai.com/blog/chatgpt

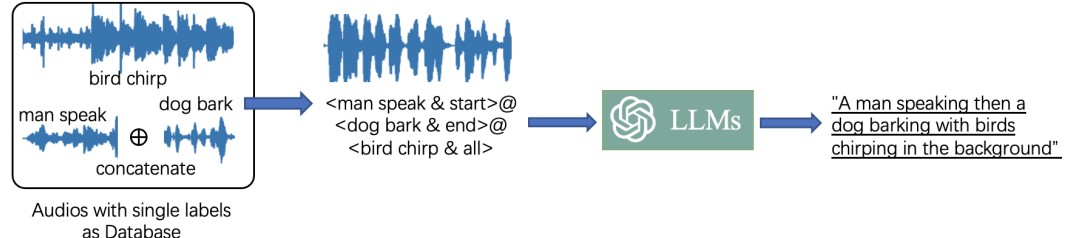

Figure 2: Overview of LLM-based data augmentation. We use single-labeled audios and their labels as a database. Composing complex audios and structured captions with these data. We then use LLM to generate diverse natural language captions by the constructed captions and appropriate prompt.

temporal information more effectively and improve the semantic alignment between the text and audio. Moreover, we present our LLM-based augmentation method, which further enhances the generalization ability and performance of our model in terms of generating audio with high semantic correspondence. In the end, we illustrate the structure of our diffusion denoiser, which is specifically designed to enhance the generation of variable-length audio.

### 3.1 OVERVIEW

Our framework overview is shown in Figure 1. Denote an audio-text pair as $(a, y)$ where $a \in R^{T_a}$ and $T_a$ is the waveform length. To mitigate the complexity of modeling long continuous waveform data, we first convert $a$ to mel-spectrogram (akin to the 1-channel 2D image) $x \in R^{C_a \times T}$, where $C_a, T \ll T_a$ denote the mel-channels and the number of frames respectively. The training process includes two stages:

1) **Training variational autoencoder**. The audio encoder $E$ takes mel-spectrogram $x$ as input and outputs compressed latent $z = E(x)$. The audio decoder $D$ reconstructs the mel-spectrogram signals $x' = D(z)$ from the compressed representation $z$. VAE solves the problem of excessive smoothing in mel-spectrogram reconstruction through adversarial training with a discriminator. The training objective is to minimize the weighted sum of reconstruction loss $\mathcal{L}_{re}$, GAN loss $\mathcal{L}_{GAN}$ and KL-penalty loss $\mathcal{L}_{KL}$.

2) **Training latent diffusion model**. Diffusion models Ho et al. (2020); Rombach et al. (2022) consists of two processes. In forward process, given the latent $z$ encoded by the VAE, diffusion model transforms $z$ into standard Gaussian distribution by $T$ steps, the data distribution of $z_t$ at step $t$ can be formulated as:

$$q\left(z_t \mid z_{t-1}\right) = \sqrt{1 - \beta_t} z_{t-1} + \sqrt{\beta_t} \epsilon_t \qquad (1)$$

Where $\beta_t \in [0, 1]$ is a predefined noise schedule hyper-parameter, $\epsilon_t \sim N(0, I)$ denotes the injected noise.

In the backward process, latent diffusion model learns to reconstruct the data distribution of $z$ with the conditional embedding $c = f_{cond}(y)$ of conditional encoder $f_{cond}$. The training objective of the diffusion module is to minimize the mean squared error in the noise space:

$$\mathcal{L}_\theta = \|\epsilon_\theta(z_t, t, c) - \epsilon\|_2^2, \qquad (2)$$

where, $\epsilon \sim \mathcal{N}(0, I)$ denotes the noise, $\epsilon_\theta$ denotes the denoising network, $t$ is the random time step. The diffusion model can be efficiently trained by optimizing ELBO Ho et al. (2020), ensuring extremely faithful reconstructions that match the ground-truth distribution.

To further improve the conditional generation performance, we adopt the classifier-free guidance Ho & Salimans (2021) technique. By jointly training a conditional and an unconditional diffusion model it can control the extent to which the condition information affects the generation at each sampling step and attain a trade-off between sample quality and diversity. At the training step, we randomly replace the audio caption with an empty string to get the empty string conditional embedding $c_\emptyset$ to train the unconditional model. During sampling, the output of the model is extrapolated further in the

direction of $\epsilon_\theta(\mathbf{z}_t, t, c)$ and away from $\epsilon_\theta(\mathbf{z}_t, t, c_\emptyset)$ with the guidance scale $s \geq 1$ :

$$\tilde{\epsilon}_\theta(z_t, t, c) = \epsilon_\theta(\mathbf{z}_t, t, c_\emptyset) + s \cdot (\epsilon_\theta(\mathbf{z}_t, t, c) - \boldsymbol{\epsilon}_\theta(\mathbf{z}_t, t, c_\emptyset)) \tag{3}$$

## 3.2 TEMPORAL ENHANCEMENT

In comparison to image data, audio data includes temporal information. A sound event can occur at any time within the audio, making audio synthesis a challenge when attempting to maintain temporal consistency. Previous approaches have encountered difficulties in dealing with captions that contain multiple sounds and complex temporal information, leading to semantic misalignment and poor temporal consistency. This can cause the generated audio to omit some sounds and produce an inaccurate temporal sequence. To address these issues, we propose the **temporal enhancement** method by parsing the original caption into structured pairs of <event & order>.

Recently, AudioGPT Huang et al. (2023b) and HuggingGPT Shen et al. (2023) take LLM (e.g., ChatGPT) as a controller to invoke other AI models for expanding LLM's capacity in addressing multi-modal tasks. Conversely, we consider the possibility of utilizing the robust language understanding capabilities of LLMs to provide temporal knowledge. Specifically, LLMs are utilized to parse the input text (the natural language audio caption) and extract structured <event & order> pairs. As illustrated in Figure 1, we use LLMs to simplify the original natural language caption and link each sound event to its corresponding order. Benefiting from enhanced temporal knowledge, the T2A model is empowered to identify sound events and corresponding temporal order. Appendix D contains further details on prompt design and additional examples of temporal information enhancement.

## 3.3 DUAL TEXT ENCODERS

To enhance the utilization of caption information, we propose a dual text encoder architecture consisting of a main text encoder CLAP Elizalde et al. (2022) that takes the original natural language caption $y$ as input, and a temporal encoder FLAN-T5 Chung et al. (2022) which takes the structured caption $y_s$ passed by LLM as input. The final conditional representation is expressed as:

$$c = Linear(Concat(f_{text}(y), f_{temp}(y_s))), \tag{4}$$

Where $f_{text}$ is the main text encoder and $f_{temp}$ is the temporal encoder. With contrastive multi-modal pre-training, the CLAP has achieved excellent zero-shot performance in several downstream tasks. We freeze the weights of the main text encoder and fine-tune the temporal encoder to capture information about the temporal order of various events. As we use LLM to parse the original natural language input, some adjectives or quantifiers may be lost in this procedure, and sometimes the structured inputs' format is incorrect. Dual text encoders can avoid information loss and are more robust in these situations. Additionally, with the frozen main text encoder, the model can maintain its generalization ability.

## 3.4 LLM-BASED DATA AUGMENTATION

A major challenge faced by the current T2A mission is the scarcity of data. While the T2I task benefits from billions of text-image pairs Schuhmann et al. (2022), there are currently only around one million open-source text-audio pairs available Huang et al. (2023a). Moreover, a significant portion of the data in this task is considered "dirty", indicating the presence of additional noises and other sounds in the audio beyond the annotated sounds. Additionally, there is a lack of data with detailed temporal annotation; many of these audios are only loosely labeled with tags instead of natural language captions. The remarkable success of the GPT series has underscored the significance of data-centric artificial intelligence Zha et al. (2023); Jakubik et al. (2022), which seeks to optimize the creation, selection, and maintenance of training and inference data to achieve optimal outcomes. To make the most effective use of the available data, we propose an LLM-based data augmentation technique. As depicted in Figure 2, we augment audio data and its corresponding text caption as follows:

- We begin by collecting data labeled with single tags to create our event database $\mathcal{D}$. This type of data is typically cleaner and less likely to contain unwanted noise or other sounds. We can then use this data to construct more complex data based on their durations.

- Then we randomly select $N \in \{2, 3\}$ samples from $\mathcal{D}$, mix and concatenate them at random. Concatenating at random intervals or overlaps ensures that the resulting audio contains temporal information. Mixing improves the models' ability to recognize and separate different sorts of audio for creating complex compositions.

- As the resulting audio is created, we synthesize structured captions based on the occurrence time and duration of each sound event by rules. For those events that appear almost throughout the audio, we bind them with "all". While for events that only partly occur in the audio, we bind them with "start", "mid" or "end" depending on the proportion of their occurrence time points in the resulting audio.

- Finally, we feed the structured captions into LLM with appropriate prompts to generate diverse natural language captions. The prompt to transform structured captions to natural language captions and some examples are displayed in Appendix D.

### 3.5 TRANSFORMER-BASED DIFFUSION DENOISER BACKBONE

Previous diffusion-based work on T2A synthesis treated the mel-spectrogram as a one-channel image similar to the approach used for T2I synthesis. However, unlike images, the mel-spectrogram is not spatially translation invariant. The height of the mel-spectrogram represents the frequency domain, which means that mel-spectrogram patches at different heights can have entirely different meanings and should not be treated equally. Furthermore, the use of a 2D-convolution layer and spatial transformer-stacked U-Net architecture limits the model's ability to generate variable-length audio. Previous works Peebles & Xie (2023); Bao et al. (2022) have shown U-Net is not necessary for diffusion network Ho et al. (2020); Rombach et al. (2022) and found transformer-based Vaswani et al. (2017) architecture as diffusion network can achieve better performance. Inspired by these works and to improve the model's ability to generate variable-length audio, we propose a modified audio VAE that uses a 1D-convolution-based model and propose a feed-forward Transformer-based diffusion denoiser backbone. While the latent of 2D-VAE audio encoder is $z = E(x) \in R^{c_e \times C_a/f \times T/f}$, where $c_e$ is the embedding dim of latent, $f$ is the downsampling rate, $C_a$ and $T$ denote the mel-channels and the number of frames of mel-spectrogram respectively, which can be seen as images' height and width. Our 1D-convolution-based audio encoder's latent is $z = E(x) \in R^{C_a/f_1 \times T/f_2}$, where $f_1, f_2$ are downsampling rates of mel-channels and frames, respectively. As the feed-forward transformer block is composed of 1D-convolution and temporal transformer, it can better understand the information of the temporal domain in latent and improves the variable length audio generation performance. Compared to the original spatial transformer, the computation complexity reduces from $O((C_a/f \times T/f)^2 \times D)$ to $O((T/f_2)^2 \times D)$, where $D$ is the embedding dimension of the transformer layer.

## 4 EXPERIMENTS

### 4.1 EXPERIMENTAL SETUP

**Dataset.** We use a combination of several datasets to train our model, including AudioCaps training set, WavCaps, AudioSet, Adobe Audition Sound Effects, Audiostock, ESC-50, FSD50K, MACS, Epidemic Sound, UrbanSound8K, WavText5Ks, TUT acoustic scene. This results in a dataset composed of **0.92 million** audio text pairs, with a total duration of approximately **3.7K hours**. More details of data preprocessing are put in Appendix A. To evaluate the performance of our models, we use the AudioCaps test set and Clotho evaluation set which contain multiple event audio samples and detailed audio captions that contain temporal information. The latter serves as a more challenging zero-shot scenario test for us, as its train set is not included in our train data.

**Evaluation methods.** We evaluate our models using objective and subjective metrics to assess the audio quality and text-audio alignment faithfulness. For objective evaluation, we include Frechet distance (FD), inception score (IS), Kullback–Leibler(KL) divergence, Frechet audio distance (FAD), and CLAP score. For subjective evaluation, we conduct crowd-sourced human evaluations with MOS (mean opinion score) to assess the audio quality, text-audio alignment faithfulness, and text-audio temporal alignment, scoring MOS-Q, MOS-F, and MOS-T respectively. More information regarding the evaluation process can be found in Appendix C.2

Table 1: The comparison between our model and baseline T2A models on the AudioCaps dataset. All the diffusion-based models run with 100 DDIM Song et al. (2020) steps for a fair comparison. Our model is tested with a classifier-free guidance scale of 5. We borrowed all the results from Liu et al. (2023); Ghosal et al. (2023) and used the model released by the authors on Huggingface to test the CLAP Score. We reimplement Make-An-Audio and replace their vocoder with our BigVGAN vocoder.

| Model | FD↓ | IS↑ | KL↓ | FAD↓ | CLAP↑ | MOS-Q↑ | MOS-F↑ | MOS-T↑ |
|---|---|---|---|---|---|---|---|---|
| GroundTruth | - | - | - | - | 0.671 | 83.5 | 82.4 | 82.8 |
| AudioGen-S | - | - | 2.09 | 3.13 | - | - | - | - |
| AudioGen-L | - | - | 1.69 | 1.82 | - | - | - | - |
| Make-An-Audio | 18.32 | 7.29 | 1.61 | 2.66 | 0.593 | 71.0 | 69.0 | 69.4 |
| AudioLDM-S | 29.48 | 6.9 | 1.97 | 2.43 | - | - | - | - |
| AudioLDM-L | 23.31 | 8.13 | 1.59 | 1.96 | 0.605 | 72.6 | 69.4 | 66.0 |
| TANGO | 26.13 | 8.23 | 1.37 | 1.87 | **0.650** | 75.2 | 72.3 | 71.4 |
| Ours | **11.45** | **11.62** | **1.25** | **1.10** | 0.641 | **79.6** | **77.6** | **77.8** |

**Baseline models.** To establish a standard for comparison, our study employs four baseline models, including Make-An-Audio Huang et al. (2023a), AudioLDM Liu et al. (2023), TANGO Ghosal et al. (2023) and AudioGen Kreuk et al. (2023). We reimplement Make-An-Audio and train it on AudioCaps Dataset. AudioLDM-S and AudioLDM-L with 454M and 1.01B parameters respectively are trained on AudioCaps, AudioSet, BBC Sound Effects, and the FreeSound dataset. TANGO is trained on AudioCaps Dataset. The close-source models AudioGen-Base and AudioGen-Large, with 285M and 1M parameters are trained on AudioCaps, AudioSet, and eight other datasets.

## 4.2 MAIN RESULTS

**Automatic objective evaluation.** The objective evaluation comparison with baseline is presented in Table 1, and we have the following observations: 1) In terms of audio quality, our model achieves better scores in FD, IS, KL, and FAD; 2) On text-audio similarity, our model presents the comparable CLAP score with TANGO; 3) Regarding temporal alignment, our model also achieves a high IS score, showing that temporal enhancement method also enhances the clarity and expressiveness of the sound events in the generated audio.

**Subjective human evaluation.** The human evaluation results show significant gains of TETA with MOS-Q of 79.6, MOS-F of 77.6, and MOS-T of 77.8, outperforming the current baselines. It indicates that raters prefer our model synthesis against baselines in terms of audio naturalness, text-audio semantics and temporal faithfulness.

**Zero-shot evaluation.** To further investigate the generalization performance of the models, we additionally test the performance of the models on the Clotho-evaluation dataset in the zero-shot scenario. Considering audios in the Clotho-evaluation dataset have different durations, we conduct two evaluations. One is generating fixed-length audios of 10 seconds, denoted as Clotho-eval-fix. The other is to generate audio that is the same length as each piece of audio in the dataset, denoted as Clotho-eval-variable. As illustrated in Table 2, our model has significantly better results than TANGO and AudioLDM-L, attributing to the scalability in terms of data usage and variable length data training.

Table 2: Comparison of our model, AudioLDM-L, and Tango on Clotho-eval datasets. The result of generating fixed-length-audios of 10 seconds is denoted as Clotho-eval-fix, and the result of generating audios the same length as Clotho-eval dataset is denoted as Clotho-eval-variable.

| Model | Clotho-eval-fix | | | | Clotho-eval-variale | | | |
|---|---|---|---|---|---|---|---|---|
| | FD↓ | IS↑ | KL↓ | FAD↓ | FD↓ | IS↑ | KL↓ | FAD↓ |
| TANGO | 32.1 | 6.77 | 2.59 | 3.61 | 36.54 | 6.65 | 2.75 | 5.23 |
| AudioLDM-L | 28.15 | 6.55 | 2.60 | 4.93 | 24.25 | 7.06 | 2.44 | 4.42 |
| Ours | **18.43** | **8.73** | **2.49** | **1.59** | **20.77** | **8.43** | **2.55** | **2.29** |

Table 3: Comparison of our model, AudioLDM-L, and Tango on Audio-caps dataset. The result of generating 5-second audios and 8-second audios are denoted as Audiocaps-5s and Audiocaps-8s respectively.

| Model | Audiocaps-5s | | | | Audiocaps-8s | | | |
|---|---|---|---|---|---|---|---|---|
| | FD↓ | IS↑ | KL↓ | FAD↓ | FD↓ | IS↑ | KL↓ | FAD↓ |
| TANGO | 31.76 | 5.50 | 2.04 | 10.53 | 18.32 | 8.39 | 1.50 | 2.04 |
| AudioLDM-L | 31.97 | 5.66 | 2.39 | 6.79 | 30.95 | 8.65 | 1.91 | 4.91 |
| Ours | **12.40** | **11.10** | **1.48** | **1.28** | **13.20** | **11.15** | **1.32** | **1.04** |

## 4.3 ANALYSES

**Variable-length generation.** Audio data can have different lengths, to investigate our models' performance on variable-length audio generation, we test to generate 5 seconds audios and 8 seconds audios on AudioCaps dataset, the results are shown in Table 3. We also test generating variable-length audios on the Clotho-eval dataset, as discussed in former paragraph. From the table, it can be seen that TANGO and AudioLDM exhibit significant performance degradation when generating audio with different lengths than the training data, as TANGO and AudioLDM pad or truncate all the training audio data to 10 seconds, and their models are based on 2D-convolution and spatial transformer to process mel-spectrogram as images. Our model maintains high performance even when generating variable-length audio samples since it is trained on audio samples of varying lengths and utilizes 1D-convolution and temporal transformers to emphasize temporal information.

Table 4: Param and diffusion module running speed comparison. The diffusion module running speed is tested on 1 A100 GPU with batch size 8 when generating 10-second audio.

| Model | Total params | Diffusion params | diffusion steps per second |
|---|---|---|---|
| AudioLDM-S | 454M | 185M | 16.59 |
| AudioLDM-L | 1.01B | 739M | 8.21 |
| Tango | 1.21B | 866M | 4.34 |
| Make-An-Audio | 453M | 160M | 24.29 |
| Ours | 937M | 160M | **37.13** |

**Model size and speed comparison.** Considering diffusion models need about one hundred steps to generate high-quality audio when using DDIM Song et al. (2020) sampling strategy, it's important to reduce the diffusion module's computation complexity. We further compare our models' diffusion module running speed and parameter quantity with other models. The results are shown in Table 4. Our model uses a relatively small diffusion module. Through the design of the feed-forward-transformer stacking structure, we are able to achieve a noteworthy acceleration while keeping the same number of parameters with Make-An-Audio. More model configuration and training details are presented in Appendix B.

Table 5: Vocoder reconstruction performance comparison on AudioCpas-test set.

| Vocoder | Model | FD↓ | IS↑ | KL↓ | FAD↓ |
|---|---|---|---|---|---|
| MelGAN | Diffsound | 26.14 | 5.4 | 1.22 | 6.24 |
| HifiGAN | Make-An-Audio | 21.79 | 5.93 | 1.03 | 6.02 |
| HifiGAN | AudioLDM | 11.45 | 8.13 | 0.22 | 1.18 |
| BigVGAN | Ours | 5.45 | 9.44 | 0.17 | 0.98 |

**Vocoder performance comparison.** The vocoder is another key component to improve the generated audio quality for mel-spectrogram-based models. We further compared the 1) BigVGAN vocoder used in our model, 2) AudioLDM's pre-trained HifiGAN, 3) Make-An-Audio's pre-trained HifiGAN and 4) Diffsound's pre-trained MelGAN. The results are shown in Table 5. We find 3) and 4) both perform worse compared with 2), while 2) and 3) use the same vocoder architecture. We find that the problem lies in the mel-processing method of 3) and 4), they use the same mel-processing method

which will result in poor performance of the vocoder. So we adopt the same mel-spectrogram extraction process as BigVGAN and get the best performance.

## 4.4 ABLATION STUDY

Table 6: The ablation study of TETA. All the models are trained on variable-length data.

| Setting | Audiocaps | | | | Clotho-eval-fix | | | |
| --- | --- | --- | --- | --- | --- | --- | --- | --- |
| | FD↓ | IS↑ | KL↓ | FAD↓ | FD↓ | IS↑ | KL↓ | FAD↓ |
| Ours | 11.45 | **11.62** | 1.25 | **1.10** | 18.43 | 8.73 | 2.49 | **1.59** |
| w/o 1d VAE + FFT diffusion | 22.69 | 5.93 | 2.17 | 3.82 | 26.59 | 6.92 | 2.67 | 6.02 |
| w/o Temporal Enhancement | 12.66 | 10.60 | 1.35 | 1.72 | 21.24 | 8.82 | 2.50 | 2.56 |
| w/o LLM Data Augmentation | **10.45** | 11.03 | **1.22** | 1.25 | 19.75 | 8.63 | **2.39** | 2.01 |
| w/o CLAP TextEncoder | 11.91 | 11.07 | 1.29 | 1.59 | **18.38** | **9.56** | 2.43 | 1.94 |

In order to assess the effectiveness of various designs in TETA, we conduct ablation studies on AudioCaps-test and Clotho-evalation set. The results are presented in Table 6. The key findings are discussed below:

**1D-VAE and FFT-diffusion** Although 2d-convolution VAE and Unet diffusion backbone has shown to perform well when trained with fixed-length data. When we tried to train them with variable-length data. They don't converge to a good result, the results even deteriorate when tested in Audiocaps set and Clotho dataset. With 1d VAE and feed-forward-transformer diffusion backbone, our model converges well when trained with variable-length data and exhibits significant advantages in generating variable-length data.

**Temporal Enhancement** The results in Tab 6 highlights the effectiveness of temporal enhancement. We use LLM to extract event and temporal information and create structured input in the form of <event & order> pairs. As this format of structured input is not in the training corpus of the text encoder, we use the trainable Flan-T5 Chung et al. (2022) as our text encoder, which leads to significant improvements in both objective scores and sound timing modeling.

**LLM Data Augementation** We use LLM Data Augmentation to further improve the model's generalization ability and alleviate the problem of high-quality data with temporal information scarcity. The absence of LLM data augmentation results in insignificant changes in the model's performance on the Audiocaps dataset. This can be attributed to we assigned higher data weights to the Audiocaps dataset, causing the model to become somewhat overfitted to this specific dataset. Conversely, when applied to the Clotho dataset, LLM data augmentation leads to a notable improvement in performance.

**Dual Text Encoder** We use the frozen CLAP encoder to extract information from the original natural language caption and trainable text encoder to extract information from the parsed input. The frozen encoder provides us with fault-tolerance mechanisms when there are information losses and errors in the parsed input while retaining some generalization capabilities. We also compare our results when the parsed input has errors on our demo page.

## 5 CONCLUSIONS

In this work, we present TETA, a temporal-enhanced T2A synthesis model. With a capable LLM to extract temporal information from the natural language caption, TETA can better understand the event order in the caption and generate semantically aligned audios. Leveraging 1D-convolutional VAE and feed-forward Transformer diffusion backbone, TETA can generate variable-length audios without performance degeneration. With complex audio reconstruction and LLM-based data augmentation, TETA is endowed with the ability to understand complex temporal relationships and combinations of multiple concepts. TETA achieves the SOTA audio generation quality in both objective and subjective metrics and extends the boundaries of the T2A. We discuss the limitations, future works, and broader impact in Appendix E.

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
