# Appendices

## TETA: Temporal-Enhanced Text-to-Audio Generation

## A   DATA DETAILS

Table 7: Statistics for the Datasets used in the paper.

| Dataset | Hours | Type | Source |
|---|---|---|---|
| Audiocaps | 109hrs | caption | Kim et al. (2019) |
| WavCaps | 2056hrs | caption | Mei et al. (2023) |
| WavText5K | 25hrs | caption | Deshmukh et al. (2022) |
| MACS | 48hrs | caption | Martín-Morató & Mesaros (2021) |
| Clothv2 | 152hrs | caption | Drossos et al. (2020) |
| Audiostock | 44hrs | caption | `https://audiostock.net` |
| epidemic sound | 220hrs | caption | `https://www.epidemicsound.com` |
| Adobe Audition Sound Effects | 26hrs | caption | `https://www.adobe.com/products/audition/offers/AdobeAuditionDLCSFX.html` |
| FSD50K | 108hrs | label | `https://annotator.freesound.org/fsd` |
| ODEON_Sound_Effects | 20hrs | label | `https://www.paramountmotion.com/odeon-sound-effects` |
| UrbanSound8K | 9hrs | label | Salamon et al. (2014) |
| ESC-50 | 3hrs | label | Piczak (2015) |
| filteraudioset | 945hrs | multi label | Gemmeke et al. (2017) |
| TUT | 13hrs | label | Mesaros et al. (2016) |

As shown in Table 7, we collect a large-scale audio-text dataset consisting of 0.92 million of audio samples with a total duration of approximately 3.7k hours. The dataset has a wide variety of sounds including music and musical instruments, sound effects, human voices, nature and living sounds, etc. For Clotho dataset, we only use its evaluation set for zero-shot testing and do not use for training. As speech and music are the dominant classes in AudioSet, we filter 95% of the samples that contain speech and music to build a more balanced dataset.

We conduct preprocessing on both text and audio data as follows:

1) We convert the sampling rate of audio to 16kHz. Prior works Yang et al. (2023); Huang et al. (2023a); Liu et al. (2023) pad or truncate the audio to a fixed length (10s), while we group audio files with similar durations together to form batches to avoid excessive padding which could potentially impair model performance and slow down the training speed. This approach also allows for improved variable-length generation performance. We truncate any audio file that exceeds 20 seconds, in order to speed up the training process.

2) We adopt the LLM-based data augmentation method in section 3.4 to construct approximately 61k additional audio-text pairs as auxiliary data.

3) For audios without natural language annotation, we apply the pseudo prompt enhancement method from Make-An-Audio Huang et al. (2023a) to construct captions aligned with the audio.

4) We assign a lower weight to the data that is not annotated with temporal information but is abundant in quantity and diversity, such as AudioSet and WavCaps data. Specifically, we traverse the AudioCaps training set and the LLM augmented data with a probability of 50%, while randomly selecting data from all other sources with a probability of 50%. For the latter dataset, we use "<text & all>" as their structured caption.

## B EXPERIMENTAL DETAILS

**Variational autoencoder.** We employed a similar VAE architecture to that of Make-An-Audio, replacing all the 2D-convolution layers with 1D-convolution layers and the spatial transformer with a temporal transformer. As detailed in Section 4.5, the output latent of VAE is $z = E(x) \in R^{C_a/f_1 \times T/f_2}$, where we choose the downsample rate of $f_1 = 4$ and $f_2 = 2$. We additionally involve R1 regularization Mescheder et al. (2018) to better stabilize the adversarial training process. We train our VAE on 8 NVIDIA A100 GPUs with a batch size of 32 and 800k training steps on AudioSet dataset. We use the Adamw optimizer Loshchilov & Hutter (2018) with a learning rate of $1.44 \times 10^{-4}$. For specific differences in hyperparameters between our VAE and that of Make-An-Audio, please see Table 8.

Table 8: Difference between Make-An-Audio VAE and our VAE

|  | Make-An-Audio VAE | TETA VAE |
| --- | --- | --- |
| Assume input tensor shape (for 10s audio) | (1,80,624) | (80,624) |
| Embed_dim | 4 | 20 |
| Convolution layer | Conv2D | Conv1D |
| Channels | 128 | 224 |
| Channel multiplier | 1,2,2,4 | 1,2,4 |
| Downsample layer position | after block 1,2 | after block 1 |
| Attention layer | spatial attention | temporal attention |
| Attention layer position | after block 3,4 | after block 3 |
| Output tensor shape | (4,10,78) | (20,312) |

**Latent diffusion.** We train our Latent Diffusion Model with on 8 NVIDIA A100 GPU with a batch size of 32 and 1.8M training steps. We use the Adam optimizer with a learning rate of $9.6 \times 10^{-5}$. For the specific hyperparameter for our latent diffusion model, please refer to Table 9.

Table 9: TETA Diffusion model backbone configurations

|  | TETA LDM |
| --- | --- |
| Input shape | (20,T) |
| Condition_embedding dim | 1024 |
| Feed-forward Transformer hidden_size | 576 |
| Feed-forward Transformer's Conv1d kernel size | 7 |
| Feed-forward Transformer's Conv1d padding | 3 |
| Number of Transformer heads | 8 |
| Number of Feed-forward Transformer block | 8 |
| Diffusion steps | 1000 |

**Model parameters of each component.** The params of each component in TETA are displayed in Table 10.

Table 10: The params of each component in TETA

| Component | Params |
| --- | --- |
| VAE | 213M |
| Diffusion Model Backbone | 160M |
| Text Encoder | 452M |
| Vocoder | 112M |
| Total | 937M |

## C EVALUATION

### C.1 SUBJECTIVE EVALUATION

To assess the generation quality, we conduct MOS (Mean Opinion Score) tests regarding audio quality, text-audio faithfulness and text-audio temporal alignment, respectively scoring MOS-Q, MOS-F, and MOS-T.

For audio quality, the raters were explicitly instructed to "focus on examining the audio quality and naturalness." The testers were presented with audio samples and their caption and asked to rate their subjective score on a 20-100 Likert scale.

For text-audio faithfulness, human raters were shown the audio and its caption and asked to respond to the question, "Does the natural language description align with the audio faithfully?" They had to choose one of the options - "completely," "mostly," or "somewhat" on a 20-100 Likert scale.

For text-audio temporal alignment, human raters were shown the audio and its caption and asked to respond to the question, "Whether the text description contains sounds time or order information. If not then select no, if yes then score based on how the audio's sound order aligns with its caption." They had to choose one of the options - "completely," "mostly," or "somewhat" on a 20-100 Likert scale. We will filter out the audio that has been selected "no" and compute MOS-T based on the remaining audio.

A small subset of the generated audio samples used in the test can be found at `https://teta2023.github.io/`.

### C.2 OBJECTIVE EVALUATION

Fréchet Audio Distance (FAD) Kilgour et al. (2018) is adapted from the Fréchet Inception Distance (FID) to the audio domain, it is a reference-free perceptual metric that measures the distance between the generated and ground truth audio distributions. FAD is used to evaluate the quality of generated audio.

The inception Score (IS) is an effective metric that evaluates both the quality and diversity of generated audio.

KL divergence is measured at a paired sample level between the generated audio and the ground truth audio, it is computed using the label distribution and is averaged as the final result.

Fréchet Distance (FD) evaluates the similarity between the generated and ground truth audio distributions. FD, KL and IS are built upon an audio classifier, PANNs Kong et al. (2020), which takes the mel-spectrogram as model input. Differently, FAD uses VGGish Hershey et al. (2017) as an audio classifier that takes raw audio waveform as model input.

CLAP score: adapted from the CLIP score Hessel et al. (2021); Radford et al. (2021) to the audio domain and is a reference-free evaluation metric to measure audio-text alignment for this work that closely correlates with human perception.

## D CHATGPT PROMPTS

The prompt templates utilized for temporal enhancement to construct structure caption from the original natural language caption and for caption data augmentation are displayed in Figure 3.

Table 11 presents some instances of the original caption and ChatGPT's outcome. For text data augmentation, we construct structured caption inputs, and Table 12 exhibits examples of such inputs and ChatGPT's corresponding outputs.

## E LIMITATIONS, FUTURE WORKS AND BROADER IMPACT

**Limitations.** TETA incorporates an additional LLM for parsing the original caption, which affects both the generation performance and running speed. Additionally, the generative diffusion model

**Prompt Template for Temporal Enhancement**

Our prompt: I want to know what sound might be in the given scene and you need to give me the results in the following format:

Question: A bird sings on the river in the morning, a cow passes by and scares away the bird.

Answer: <running water & all>@<birds chriping & start>@<cow footsteps & mid>@<birds flying away & end>

Question: cellphone ringing a variety of tones followed by a loud explosion and fire crackling as a truck engine runs idle.

Answer: <variety cellphone ringing tones & start>@<loud explosion & end>@<fire crackling & end>@<truck engine idle & end>

Question: Train passing followed by short honks three times.

Answer: <train passing & all>@<short honks three times & end>

All indicates the sound exists in the whole scene, Start, mid, end indicates the time period the sound appear.

Question: A metal clank followed by motor vibrating and rumbling.
Answer:_________

LLMs

<metal clank & start>@<motor vibrating & mid>@<motor rumbling & end>

**Prompt Template for Data Augmentation**

Our prompt: I want you to generate the text describing the sound scene based on the structured input:

Question: <running water & all>@<birds chriping & start>@<cow footsteps & mid>@<birds flying away & end>

Answer: A bird sings on the river in the morning, a cow passes by and scares away the bird.

Question: <variety cellphone ringing tones & start>@<loud explosion & end>@<fire crackling & end>@<truck engine idle & end>

Answer: cellphone ringing a variety of tones followed by a loud explosion and fire crackling as a truck engine runs idle.

Question: <Applause Clapping & all>@<Gong & start>@<Steel guitar and slide guitar & mid>

Answer: Gong sounds start followed by steel guitar and slide guitar with applause and clapping in the background

All indicates that sound exists in the whole scene. Start, mid, end indicates the time period the sound appear. Please answer the following questions, each answer should be start from a newline:
1. <Accelerating & start>@<Heavy engine low frequency & mid>
2. <Hammer & all>@<Clatter & all>@<air conditioner & start>

LLMs

1. The sound of accelerating starts followed by the heavy engine's low frequency in the middle.
2. The sound of a hammer and clatter can be heard throughout, accompanied by the hum of an air conditioner starting up.

Figure 3: The prompt templates we used for temporal enhancement and data augmentation. We use the symbol '&' to split the sound event and the time order. We use the symbol '@' to split <event & order> pairs.

Table 11: Examples of using ChatGPT for temporal enhancement from AudioCaps trainset

| Natural language input | ChatGPT's output |
|---|---|
| A woman talks nearby as water pours | <woman talking& all>@<water pouring& all> |
| Two men exchange words, then a car engine revs followed by a siren and fade in music | <two men talking& all>@<car engine revving& start>@<siren& mid>@<music fading in& end> |
| A crowd is cheering and shouting, thumping occurs, an adult female speaks, and an adult male speaks | <crowd cheering and shouting& all>@<thumping& start>@<adult female speaking& mid>@<adult male speaking& end> |

employed by TETA requires multiple iterative refinements for synthesis, which can be time-consuming to produce high-quality results. Furthermore, the speech in the generated audio can be intelligible.

Table 12: Examples of using ChatGPT for data augmentation

| Structured input | ChatGPT's output |
|---|---|
| <bark howl dog& start>@<typing Typewriter& mid>@<breathing& end> | A dog barks and howls while someone types on a typewriter, then the sound of breathing takes over |
| <crowing& all>@<car& all>@<female singing& end> | A rooster crows as a car passes by, and the scene ends with the sound of a female singing |
| <sneezing& all>@<bicycle bell ring& start>@<typewriter & end> | The sound of sneezing is heard throughout, with a bicycle bell ringing at the start and the sound of a typewriter at the end |

**Future works.**  We leave the T2A system which supports speech synthesis for future work. As we have seen great potential in our LLM-based data augmentation, with elaborate prompts and merge rules, it can be used to merge speech, singing, sound events, and music to create a more universal audio scenario. Enabling the training of a model that can generate universal audios with meaningful speech and music with ideal melody. In addition, we aim to implement T2A systems that could take structured inputs as optional auxiliary inputs instead of required inputs.

**Broader impacts.**  At the same time, we acknowledge that TETA may lead to unintended consequences such as increased unemployment for individuals in related fields such as sound engineering and radio hosting. Furthermore, there are potential concerns regarding the ethics of non-consensual voice cloning or the creation of fake media.