# OpenReview forum: "TETA: Temporal-Enhanced Text-to-Audio Generation"
_ICLR.cc/2024/Conference — ICLR 2024 Conference Withdrawn Submission_

### Official Review · Reviewer_NGeL · 2023-10-26

**Soundness:** 2 fair
**Presentation:** 2 fair
**Contribution:** 2 fair
**Rating:** 3
**Confidence:** 4

**Summary:**

* The paper’s main contribution is a technique to train text-to-audio models such that the temporal order constraints mentioned in the input text is adhered to in the generated audio.
* The proposed method models temporal ordering constraints by using an LLM like ChatGPT to extract relationships in the form of <event, order> pair from the original input text prompt. It then uses two separate text encoders, one for the original input text and one for the extracted structured pairs. The two representations are concatenated and used as a conditioning input to the diffusion denoiser.
* The paper also proposes a data synthesis method that produces paired audio-text samples that are temporally aligned in the audio and text content. Audio samples from other datasets that have single event labels as tags are mixed and concatenated using simple templates. A corresponding diverse audio caption is also synthesized using an LLM by feeding it the individual event information.
* A secondary contribution is a 1D CNN based audio encoder and Transformer based Diffusion model that is supposed to provide better results on variable length generation.
* Experimental results on Audiocaps and Clotho show that the proposed method can provide better numbers on the metrics meant for evaluating audio generation.

**Strengths:**

* The paper brings attention to an important problem where the audio generation models lack full semantic consistency with the input text prompt. The proposed method particularly addresses the temporal order semantics by leveraging the capability of LLMs to simplify the problem for the audio generation model itself.

**Weaknesses:**

* The evaluation metrics used in the paper are not capable of measuring the temporal consistency that the proposed method is trying to improve. It is not clear why the IS metric is suitable for measuring temporal alignment.
* The experimental results provides an analysis of the proposed method as any other general audio generation method. This is necessary but not sufficient to justify the claims on improving temporal consistency.
* The section on variable length generation, 1D CNN based audio encoding and Transformer diffusion model is very confusing.
* The 1D CNN’s latent representation is still a 2D tensor where there is subsampling in the mel-channels dimension. The only difference is that it is a single channel 2D tensor. A 1D CNN’s output should typically be a 1D vector for a single channel.
* It is not clear how the combination of 1D CNN audio encoding and Transformer diffuser is better for variable length generation i.e. what aspect of it is making it suitable for variable length output. The model doesn’t seem to be an autoregressive model, hence, it is still trained and inferred with fixed lengths.

**Questions:**

* Minor fix: The abbreviation FFT in Fig. 1 is overloaded. It can be confused for the more commonly known Fast Fourier Transform. However, I am guessing the authors are referring to the Feed Forward Transformer.

---

### Official Review · Reviewer_UN3N · 2023-10-27

**Soundness:** 3 good
**Presentation:** 1 poor
**Contribution:** 2 fair
**Rating:** 3
**Confidence:** 3

**Summary:**

The authors propose a framework for diffusion text-to-audio synthesis with the following contributions: They improve the diffusion process by augmenting the input text with structured <event & order> pairs. They propose a transformer model for variable-length audio. They create new datasets by mixing and concatenating audio clips and augmenting their labels to natural text using LLMs.

**Strengths:**

- The authors address temporal disorder, which is an important issue with current text-to-audio diffusion models.
- The authors also address an important aspect of processing audio spectrograms: that mel-spectrograms are not translation-invariant on the frequency axis.
- The authors compare with many models and present an ablation study.

**Weaknesses:**

1. The author discusses the challenges of generating audio clips of variable length. For longer audio clips, it is not clear for me why they opt for a structure that only supports 3 temporal positions: start, end, and all.
2. This sentence is not clear: "Additionally, 2D spatial structures widely used in T2A works lead to unsatisfactory audio quality when generating variable-length audio samples since they do not adequately prioritize temporal information." What do you mean by "they do not adequately prioritize temporal information"? and I don't understand how it is related to "variable-length" audio.
3. Reproducibility: Although the method to generate the dataset is clearly described, The subsystems are not clearly described, which makes reproducing the result very hard, especially in Section 3.5  (see the next point).
4. Section 3.5 is not clear at all:
   1. The 1D-convolution-based audio encoder’s latent is  of the shape C_a/f_1 X T/f_2 . It's not clear how  the 1D convolution results in
     this shape, is there no channels dimension, or are you just pooling over time and frequency?
   2. The complexity calculation is not clear: "Compared to the original spatial transformer, the computation complexity reduces from O( (TXF)^2 x D ) to O(T^2 X D)". It's not clear what happened to the frequency dimension; was it merged into the latent features D? I also think the related work uses only U-nets so to what models are you comparing exactly here?
5. Evaluation:
   1.  Since the main motivation of the paper is temporal disorder, I think there should be a comparison with LM-based audio generation (such as AudioLM etc...) where the LM can resolve temporal dependencies.
   2. Again, since this work addresses mainly temporal disorders, I believe that there should be a clear evaluation of this exact point.
   3. Can you please explain the evaluation metrics in Section 4.1 clearly (directly) and what they mean when evaluating temporal disorder. I think in its current form, it's ambigious. Additionally, make sure you're using the same evaluation methods (AudioLDM uses a different classifier than AudioGen to calculate the KL).

**Questions:**

Could you kindly address each of the above concerns point-by-point and clear any misunderstandings

---

### Official Review · Reviewer_Kz7t · 2023-10-31

**Soundness:** 2 fair
**Presentation:** 2 fair
**Contribution:** 2 fair
**Rating:** 5
**Confidence:** 4

**Summary:**

This paper proposes a new method for generating temporal-accurate events in text-to-audio generation models. This paper proposes several changes in addition to the usual text-to-audio paradigm in certain ways. First, an LLM is used to generate labels of even & order pairs, and a data augmentation is used to concatenate audio snippets and generate corresponding text for that. Then, a temporal encoder is used to provide more alignment information. Last, a transformer-based backbone is used instead of conventional UNet.

**Strengths:**

The paper presents an enriched paradigm for text-to-audio generation that not only produces audio but also ensures better alignment and ordering of generated events. This is achieved by integrating event order labeling through LLM and the inclusion of a temporal encoder.

**Weaknesses:**

1. Figure 1 might benefit from clearer representation, especially regarding the system's overall structure. Specifically, the diffusion model could be labeled more prominently as that is the core model being trained. It would also be beneficial to clarify the relationship between the "FFT block x N" and the diffusion model, considering they might correspond to the same denoising step as noted in the text (`denoising p_{\theta}(x_{t-1}| x)0`).
2. The LLM-based data augmentation approach bears some resemblance to the approach presented in the "make-an-audio" paper [1]. It might be valuable to highlight what differentiates the proposed method from the one cited.
3. In Section 4.4's Table 6, the 1D VAE and FFT diffusion seem to be the primary contributors to the observed performance enhancement. It might be worth discussing how the other combined design elements factor into this performance boost.
[1] Huang, R., Huang, J., Yang, D., Ren, Y., Liu, L., Li, M., ... & Zhao, Z. (2023). Make-an-audio: Text-to-audio generation with prompt-enhanced diffusion models. arXiv preprint arXiv:2301.12661.

**Questions:**

Table 5 suggests that the vocoder design plays a significant role in the generation's performance. Given that the vocoder is not the paper's primary focus, I'm curious if the authors considered using the proposed diffusion model alongside the same vocoder found in baseline models, providing a more controlled comparison.

---

### Official Review · Reviewer_J54j · 2023-11-03

**Soundness:** 2 fair
**Presentation:** 3 good
**Contribution:** 2 fair
**Rating:** 3
**Confidence:** 4

**Summary:**

This paper proposes a new system for text-to-audio generation that offers better temporal and compositional capabilities. The main novelty of the system is the temporal encoder that uses an LLM and a temporal-enhanced representation. The experimental results show superior results of the proposed system over several baseline models.

**Strengths:**

- The proposed temporal-enhanced representation is clever. Although its scalability is unclear, this work presents the first attempt to achieve interpretable temporal-aware text-to-audio generation system. One future direction is to extend this to a representation similar to WavJourney where we can have fine-grained temporal resolution over the sound events.
- Extensive ablation study help understand the necessity of each component of the proposed system.

**Weaknesses:**

- While the authors compare the proposed system with many baseline models, most baseline models are not designed for temporally-compositional text-to-audio synthesis except Make-An-Audio. However, the reimplementation of Make-An-Audio seems to miss many critical components. For example, Make-An-Audio proposed a augmentation method to approach the same temporally-compositional problem addressed in this paper. Moreover, the reimplementation was trained on only AudioCaps, while the proposed system was trained on a much larger collection of dataset, making the comparison unfair. It remains unclear how the proposed method compared to the approaches proposed in Make-An-Audio.
- The authors claim state-of-the-art performance for text-to-audio synthesis while the proposed systems are only evaluated on AudioCaps and Clotho, which are both rather noisy. Moreover, as described in Section 4.4, the authors assign larger weight to AudioCaps during training, making it more likely to overfit on AudioCaps. It remains unclear whether the proposed system can work well on other types of inputs.

**Questions:**

- (Abstract) "2D spatial structures" -> While this term has been used intensively in this work, I don't know what it refers to actually. Do you mean the 2D time-frequency representation? Please clarify this.
- (Figure 2) "diverse" -> Did you measure the diversity of the generated captions? How did you measure that?
- (Section 3.5) "Compared to the original spatial transformer, ..." -> Not sure what this refers to. Isn't the original model convolution-based?
- (Section 4.1) What's the difference between MOS-F and MOS-T?
- (Table 1) What's the standard errors for the MOS? It's hard to make any conclusions without error bars. Also, were all these baseline models trained on AudioCaps?
- (Section 4.1) "We reimplement Make-An-Audio and train it on AudioCaps Dataset." -> I don't believe this is how Make-An-Audio was trained? Did you apply the augmentation technique proposed in Make-An-Audio? Also, it becomes a unfair comparison when your model is trained on a larger dataset than Make-An-Audio. In my understanding, Make-An-Audio is the most relevant system that we should compare TETA to, so please clarify how you reimplemented Make-An-Audio for a fair comparison.
- (Section 4.2) I didn't get the idea of using IS to measure temporal alignment. Is this a standard metric to measure temporal alignment? What's the intuition and rationale behind this?
- (Section 4.2) What's the mean/median/stddev of audio length for Clotho-eval-variable? Some statistics of the length will be helpful here. It's unclear to me whether this means generating shorter or longer audio.
- (Table 2 & 3) Should compare with Make-An-Audio. These models are not built and trained for temporally compositional audio generation.
- (Section 4.3) What are the vocoders used in the comparison in Table 1? How can we disentangle the effects of vocoders from the synthesis model? Are FD values comparable among models -- are they using the same parameters for mel spectrogram computation?
- (Section 4.4) When we tried to train them with variable-length data. They don't converge to a good result, the results even deteriorate when tested in AudioCaps and Clotho dataset.

Here are some other comments and suggestions:

- The format of citations are confusing. Please use parentheses when appropriate. For example, the citations in the first line of introduction should be parenthesized.
- (Section 3.1) "ensuring extremely faithful..." -> "extremely" is not a proper word to be used in a scientific paper
- (Section 3.4) "Then we randomly select N ∈ {2, 3} samples from D, mix and concatenate them at random." -> One of the problems of such artificially created mixture is the inconsistency of sound characteristics in different component sounds such as reverberations and microphone quality.
- (Section 3.4) "we bind them with start, mid or end depending ..." -> This offers a poor time resolution. It remains unclear how this method can be scaled up to long-form audio generation.
- (Section 3.5) "Furthermore, the use of a 2D-convolution layer and spatial transformer-stacked U-Net architecture limits the model's ability to generate variable-length
audio." -> Not sure why this is the case. I believe UNets can generate the desired output length if you can provide a noise sequence of the same length. Please provide more supporting evidence for your argument.
- (Section 4.1) "This results in a dataset composed of 0.92 million audio text pairs, ..." -> From this I assume there is an intense filtering involved during the preprocessing since AudioSet has 2M samples and is rather noisy. Please include some descriptions on the data cleaning approach in the main text as it is  a critical step.
- (Section 4.2) AudioCaps and Clotho are both noisy datasets, so FD and FAD doesn't necessarily measure the audio quality. Moreover, I believe KL doesn't measure the audio quality. CLAP is also known to perform poorly on complex audio with multiple sources and temporal ordering.
- (Section 5) "TETA achieves the SOTA audio generation quality in both objective and subjective metrics and extends the boundaries of the T2A." -> This is a problematic statement as the authors only evaluated the systems on AudioCaps and Clotho.